# Long-Term Enclosure Can Benefit Grassland Community Stability on the Loess Plateau of China



**Jing Liu** [1,2,†] , **Xue Yang** [1,2,†] , **Hossein Ghanizadeh** [3] , **Qian Guo** [1] , **Yongming Fan** [1] , **Bo Zhang** [4,5] , **Xinhui Yan** [1,2] , **Zhongming Wen** [1,2] and **Wei Li** [2,4,*]

1 College of Grassland Agriculture, Northwest A&F University, Yangling 712100, China; lj15848230276@nwafu.edu.cn (J.L.); xyang2019@nwafu.edu.cn (X.Y.); Guoqian18@nwafu.edu.cn (Q.G.); fanyongming@nwafu.edu.cn (Y.F.); yanxinhui@nwafu.edu.cn (X.Y.); zmwen@ms.iswc.ac.cn (Z.W.)
2 Institute of Soil and Water Conversion, Northwest A&F University, Yangling 712100, China
3 School of Agriculture and Environment, Massey University, Palmerston North 4442, New Zealand; H.GhaniZadeh@massey.ac.nz
4 Institute of Soil and Water Conservation, Chinese Academy of Sciences and Ministry of Water Resources, Yangling, Shaanxi 712100, China; zhangbo19@mails.ucas.ac.cn
5 University of Chinese Academy of Sciences, Beijing 100049, China
* Correspondence: liwei2013@nwsuaf.edu.cn
† These authors contributed equally to this work.

**Abstract:** Fertilization and grazing are two common anthropogenic disturbances that can lead to unprecedented changes in biodiversity and ecological stability of grassland ecosystems. A few studies, however, have explored the effects of fertilization and grazing on community stability and the underlying mechanisms. We conducted a six-year field experiment to assess the influence of nitrogen (N) fertilization and grazing on the community stability in a long-term enclosure and grazing grassland ecosystems on the Loess Plateau. A structural equation modeling method was used to evaluate how fertilization and grazing altered community stability. Our results indicated that the community stability decreased in the enclosure and grazing grassland ecosystems with the addition of N. The community stability began to decline significantly at 4.68 and 9.36 N g m$^{-2}$ year$^{-1}$ for the grazing and enclosure grassland ecosystems, respectively. We also found that the addition of N reduced the community stability through decreasing species richness, but a long-term enclosure can alleviate its negative effect. Overall, species diversity can be a useful predictor of the stability of ecosystems confronted with disturbances. Also, our results showed that long-term enclosure was an effective grassland management practice to ensure community stability on the Loess Plateau of China.

**Keywords:** community stability; disturbances; diversity effect; grazing

## 1. Introduction

Community stability refers to the ability of a community to maintain species compositions and productivity over time, and to recover former levels of productivity or species compositions after a disturbance [1]. The perturbation of grassland community stability can adversely influence the sustainable function of grassland ecosystems [1,2]. The community stability of grassland ecosystems can be perturbed by several natural and anthropogenic causes [2]. Grazing and fertilization are the most common anthropogenic disturbances that can lead to alterations in biodiversity and community stability in grassland ecosystems [3,4]. The excessive application of fertilizer can lead to nutrient enrichment, which can affect grassland ecosystem functioning [4,5]. Some studies found that community stability decreased with increasing the level of N fertilization [1,4,6], but a few other studies found that long-term N fertilization increased community stability via enhancing species dominance [7]. Grazing has been found to decrease community stability [8,9], while long-term enclosure can facilitate vegetation recovery and increase plant productivity [10]. Other

studies, however, have shown that grazing is beneficial to community stability compared to long-term enclosure [11,12], because grazing enhances the nutrient cycling between plants and soil, and also regulates the relationships among plant species in ecosystems [13–15].

Understanding the mechanisms underlying changes in community stability is necessary to predict how ecosystems work in response to perturbations [6,16]. According to Grman et al. [6], there are four potential mechanisms by which perturbations can affect the stability of ecosystems: (1) Perturbations can reduce community stability via a reduction in species diversity [1,7,17,18]. In this regard, several studies indicated that communities with higher species diversity tended to be more stable; thus, more tolerant to perturbations [19,20]. (2) Perturbations may reduce community stability by reducing the strength of compensatory dynamics [1]. Compensatory dynamics is a function of negatively correlated interactions among species in a community. Compensatory dynamics describes a balancing process where a decline in the abundance of a species is compensated for by an increase in the abundance of similar species. Compensatory dynamics can act as a mechanism to promote stability; thus, it can play a crucial role in sustaining ecosystem stability [6,21]. Perturbations may interfere with the community-wide synchrony of species and result in a reduction in the strength of compensatory dynamics [6,22] and a decline in community stability [1]. (3) Perturbations may reduce community stability via changes in the pattern of fluctuations in the abundance of species through time and space [6]. The scaling relationship between the mean abundance and temporal variance in species abundance (mean–variance scaling relationship) was first described by Taylor et al. [23]. The slope of this scaling relationship determines community stability [24–26]. Hence, if perturbations change the slope of the scaling relationship of mean–variance, community stability will alter [3,27]. (4) Perturbations may result in the removal of dominant species from a community. Dominant species play a pivotal role in stabilizing communities [3,27,28]. Hence, the removal of dominant species by perturbations can negatively affect the stability of community [29].

Unfavorable factors such as global warming and human perturbation have led to destructive effects on community stability [30]. Enclosure has been the one of the most important management efforts to restore the community stability in grassland ecosystems [31,32]. Hence, in order to maintain ecosystem balance and improve ecosystem production, China has implemented a series of ecological restoration projects such as the Grain-for-Green program through converting croplands into grasslands, shrublands or forests since 1999, and the Returning Grazing Lands to Grasslands program through grazing exclusion since 2003 [33–35]. However, grazing exclusion may not always have beneficial impacts on species diversity and grassland ecosystem stability [36]. Thus, a better understanding of the response of community stability to perturbations is critical for projecting the impacts of future global change scenarios. While several studies have shown that N fertilization and grazing may alter ecosystem stability [3,4,17,37], the results have not been consistent. In addition, among mechanisms underlying changes in community stability, the mechanisms associated with the community stability in grassland ecosystems when facing two anthropogenic disturbances, grazing and fertilization, are unknown. To examine how the addition of N fertilizer and grazing affect community stability, and to investigate the underlying mechanisms, we analyzed six years data (2013–2018) collected from long-term N fertilization and grazing experiments conducted on the Loess Plateau. Our study aimed to test the following hypotheses: (1) the addition of N fertilizer and grazing can decrease community stability, (2) long-term enclosure can be conducive to community stability, and (3) under the addition of N fertilizer and grazing, community stability can be positively correlated with species diversity.

## 2. Material and Methods

### 2.1. Site Description

This study was conducted in the steppe grasslands of Yunwu Mountain National Natural Reserve located on the Loess Plateau (106°21′–106°27′ E, 36°10′–36°17′ N), 1800–

2150 m a.s.l., Ningxia, China. The average annual temperature in this area is 7.01 °C, and the mean annual precipitation is 425 mm, with 60–75% of the precipitation falling from July to September [38]. There are more than 297 plant species in the area, but the main species are *Stipa bungeana*, *Stipa grandis*, *Thymus mongolicus* and *Artemisia sacrorum* [39]. The soil in the study area is montane gray-cinnamon soil classified as a Calci-Orthic Aridisol according to the Chinese taxonomic system, which is equivalent to a HaplicCalcisol in the FAO/UNESCO system [40].

### 2.2. Experimental Design

We used a split-plot design with six levels of N fertilization and six blocks nested within two grazing treatments (ungrazed [long-term enclosure], grazed). We established 72 permanent 4 m × 6 m plots, with 36 plots placed within a long-term enclosure grassland, and the remaining 36 plots placed within a grazing grassland. For both the grazed and ungrazed treatments, the plots were randomly arranged in a regular six-by-six matrix, with 2-m buffer strips between plots. The N fertilization was at 0, 5, 10, 20, 40, 80 g m$^{-2}$ year$^{-1}$ (hereinafter referred to as N0, N5, N10, N20, N40 and N80), corresponding to 0, 2.34, 4.68, 9.36, 18.72, 37.44 g N m$^{-2}$ year$^{-1}$, respectively. The enclosure grassland had been fenced for 38 years while the grazing grassland had been grazed for 38 years. Grazing was performed once a month and lasted for 10 consecutive days. The grazing period was from June to August, and the grazing intensity was three sheep ha$^{-2}$. The sheep were allowed to graze freely from 8:00 a.m. to 6:00 p.m. each day. The N fertilization was applied in the form of $CO(NH_2)_2$ fertilizer annually from 2013 at the beginning of the growing season (usually at the end of April), and the fertilizer was applied before rain to avoid the need for watering.

### 2.3. Plant Community Composition Monitoring

Within each plot, a permanent quadrat of 0.5 m × 0.5 m was randomly designated and placed at least 0.5 m from the margin to avoid edge effect. From 2013 to 2018, species richness, species abundance and plant height were determined in each permanent quadrat during mid-to-late August. Species richness was measured as the number of species in the plant community. The number of individuals per species was recorded as species abundance. Plant height was measured by recording the height of five randomly selected plants of each species and averaging the mean.

To investigate the response of plant communities from a functional group perspective over time following N addition and grazing, we divided vegetation into three functional groups, namely, grasses (including sedge species), legumes and forbs, according to Stöcklin and Körner [41]. The research method based on functional groups is an effective research approach to assess the response of plant communities to environmental changes [42,43].

### 2.4. Plant Aboveground Productivity Measurement

From 2013 to 2018, during mid-to-late August, the aboveground material of all species within 0.5 m × 0.5 m quadrats randomly located in each plot was clipped at ground level and put into envelopes according to species classification. Dead material was also included. The samples were then dried to a constant weight at 80 °C and weighed. Care was taken not to resample in previously clipped areas when sampling to protect ecological balance.

### 2.5. Relationships between Community Stability and the Underlying Mechanisms

The temporal stability of aboveground biomass over a six-year period from 2013 to 2018 in the grassland ecosystem was estimated using the Equation (1) [44,45]:

$$S = \mu/\eth \tag{1}$$

where *S* is temporal stability of the ecosystem, $\mu$ is the mean of aboveground biomass, and $\eth$ is the standard deviation of aboveground biomass over time. In order to remove the

effects of the trend from the data set (i.e., detrending), the six-year data were divided into non-overlapping intervals of shorter duration (e.g., two or three years) as suggested by Tilman et al. [45].

To measure the compensatory dynamics among species, the synchrony of community-wide species was estimated using Equation (2), as suggested by Isbell et al. [25]:

$$\varphi b = \sigma^2_{bT} / \left( \sum_{i=1}^{S} \sigma_{bi} \right)^2 \tag{2}$$

where $\varphi b$ is the community-wide synchrony of species based on species biomass, $\sigma^2_{bT}$ is the variance in community biomass, $\sigma_{bi}$ is the standard deviation in the biomass of species $i$, and $S$ is the number of species in the community [7]. A similar method was used to investigate the compensatory dynamics among functional groups. The $\varphi b$ value of 1 indicates that there is perfect synchrony among species/functional groups (i.e., no significant compensatory dynamics), while the $\varphi b$ value smaller than 1 indicates that there is significant compensatory dynamics among species/functional groups [6,16].

Equation (3) was used to investigate if there is evidence for mean–variance scaling, as suggested by Lehman and Tilman [24].

$$\sigma^2 = c\mu^z \tag{3}$$

where $\sigma^2$ is the temporal variance in the biomass of a species, $\mu$ is the mean biomass of the species, $c$ is a constant and $z$ is the scaling power. The parameter $z$ represents the slope of the mean–variance relationship for each permanent quadrat [24]. The $z$ value greater than 1 indicates that the mean–variance scaling contributes to ecosystem stability [17,24].

The role of dominant species in community stability was evaluated using Simpson's dominance index (Equation (4)) [46]:

$$D = 1 - \sum_{i=1}^{S} Pi^2 \tag{4}$$

where $D$ is the Simpson's dominance index, $S$ is the number of species, and $Pi$ is the biomass proportion of species $i$ [1,7].

### 2.6. Statistical Analysis

All statistical analyses were performed using Excel 2010, SPSS 22.0 and SPSS Amos 25.0, at $\alpha = 0.05$. Shapiro–Wilk and Bartlett Shapiro–Wilk tests were used to assess normality and homogeneity of variance, respectively. The effects of N fertilization and grazing on total aboveground biomass, functional groups biomass, species richness, Simpson's dominance index data were statistically analyzed using a two-way repeated-measure ANOVA, with year as the within-subject effect, and fertilization and grazing as the fixed between-subject effects. Tukey tests were used to determine the statistical differences in the mean values, and to assess the effect of year, grazing, fertilization and their interactions on the mean values. The effects of N and grazing on the temporal stability of community biomass and functional groups biomass, species asynchrony and functional groups asynchrony were also statistically analyzed using a two-way ANOVA, and the mean values were compared using Tukey tests. The relationships between community stability and the underlying mechanisms (i.e., species diversity, compensatory dynamics, mean–variance and species dominance) were assessed using simple linear regression. The scaling power coefficient z was determined using a two-way ANOVA.

Structural equation modeling (SEM) is a statistical method to analyze the relationship between variables based on the covariance matrix of variables [47]. The SEM method was used to evaluate how fertilization and grazing altered community stability through factors

affecting community stability considered in the linear regression. The final model was attained after sequentially eliminating non-significant pathways.

## 3. Results

### 3.1. Effects of N Fertilization and Grazing on Community Stability

Fertilization and grazing had a significant impact on community stability (Figure 1, Table A1). There was a general decline in the community stability with increasing levels of N in the enclosure and grazing grassland ecosystems. However, a significant reduction in the community stability was recorded at N levels greater than 4.68 N $m^{-2}$ $yr^{-1}$ ($N_{20}$ to $N_{80}$) in the grazing grassland ecosystem while a significant reduction in the community stability in the enclosure grassland ecosystem was observed at N levels greater than 9.36 N $m^{-2}$ $yr^{-1}$ ($N_{40}$ to $N_{80}$) (Figure 2a). The results also demonstrated that there were significant interactive effects of N versus grazing on the community stability in the grassland ecosystems (Table A1).

Similarly, fertilization and grazing affected the stability of functional groups. The stability of functional groups was also found to decline with increasing the level of N in the enclosure and grazing grassland ecosystems (Figure 2b–d). However, differential responses to N addition were recorded among functional groups. Fertilization significantly affected the stability of forbs and grasses but not legumes, while grazing significantly affected the stability of forbs only (Table A1, Figure 2b–d). The results also showed that there were, however, no significant interactive effects of N versus grazing on the stability of all species across functional groups (grasses, legumes and forbs) in the grassland ecosystems (Table A1).

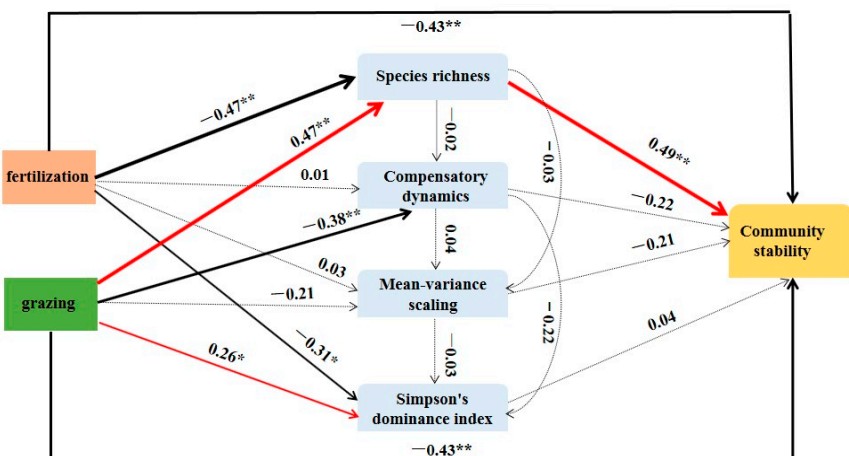

**Figure 1.** A structural equation model of the effects of the anthropogenic disturbances, N addition and grazing on the community stability in the grasslands on the Loess Plateau of China. The structural equation model considered all possible pathways through which N addition and grazing can affect community stability. Significant positive and negative pathways are shown using red and black arrows, respectively. Non-significant pathways are indicated by gray dashed arrows. Arrow width is proportional to the strength of the relationship. Bold numbers represent the standard path coefficients. * = $p < 0.05$, ** = $p < 0.01$.

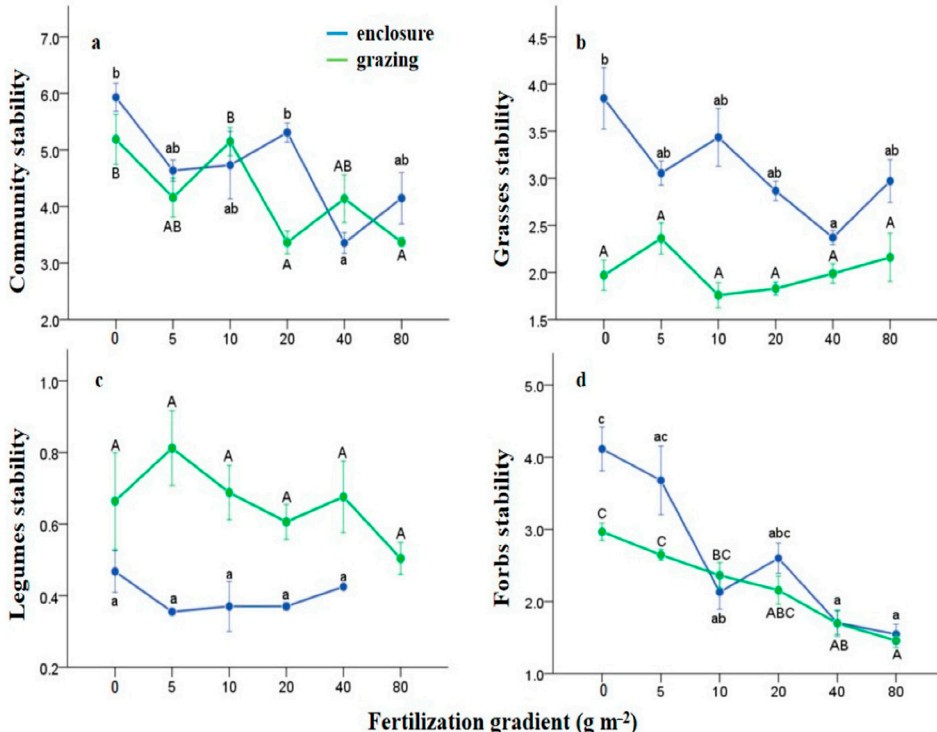

**Figure 2.** The response of community stability (**a**) and the stability of species across functional groups (grasses (**b**), legumes (**c**) and forbs (**d**)) to increasing levels of N in the enclosure and grazing grassland ecosystems; the green and blue lines represent the grazing and enclosure grassland ecosystems, respectively. Different lower-case and upper-case letters indicate significant differences between mean values within the data points for the enclosure grassland ecosystem and grazing grassland ecosystem, respectively, according to Tukey's tests at 5% probability.

There was a significantly positive relationship between the stability of all species across functional groups (grasses, legumes and forbs) and community stability (Figure A1), indicating that under the conditions of N addition and grazing, the stability of functional groups contributed to community stability. Similarly, our results showed that population stability played a crucial role in community stability. For instance, species such as *Stipa przewalskyi*, *Carex aridula*, *Artemisia sacroru*, *Thymus mongolicus*, *Dendranthema lavandulifolium* had a certain degree of stability when facing N addition and grazing (Table A2).

*3.2. Effects of N Fertilization and Grazing on Species Diversity*

N addition and grazing had a significant impact on species richness; however, there were no interactive effects of N versus grazing on species richness (Table 1). The addition of N resulted in differential impacts on the richness of species across functional groups, as the addition of N had only a significant effect on the richness of forbs (Table 1, Figure 3). However, grazing appeared to have significant impacts on all species across functional groups, while there was only a significant interactive effect of N versus grazing on the richness of forbs (Table 1). It was also noted that both species richness and functional groups richness varied significantly with years, and there were significant interactive effects of grazing versus year on both species richness and functional groups richness (Table 1, Figure A2). While no significant N versus year interactive effects on grasses richness were recorded, the N versus year interaction was significant for species richness, legumes richness and forbs richness (Table 1, Figure A2). Species richness and functional groups richness decreased with increasing the level of N in the enclosure and grazing grassland ecosystems (Figure 3a). However, species richness and functional groups richness in the grazing grassland ecosystem were greater than those of the enclosure grassland ecosystem (Figure 3). Increasing the level of N did not have a significant impact on grasses richness

in the enclosure and grazing grassland ecosystems (Figure 3b). Legumes richness only significantly reduced with increasing the level of N in the enclosure grassland ecosystem (Figure 3c). The richness of forbs significantly reduced with increasing the level of N in the enclosure and grazing grassland ecosystems (Figure 3d).

**Table 1.** The results of two-way repeated-measure ANOVAs of effects of year, nitrogen, grazing and their interaction on species richness.

| | Y | | | N | | | G | | | Y*N | | | Y*G | | | Y*G*N | | | N*G | | |
|---|---|---|---|---|---|---|---|---|---|---|---|---|---|---|---|---|---|---|---|---|---|
| | df | F | *p* | df | F | *p* | df | F | *p* | df | F | *p* | df | F | *p* | df | F | *p* | df | F | *p* |
| SR | 5 | 69.46 | 0.000 ** | 5 | 21.96 | 0.000 ** | 1 | 57.94 | 0.000 ** | 25 | 1.97 | 0.005 ** | 5 | 6.67 | 0.000 ** | 25 | 2.30 | 0.001 ** | 5 | 1.83 | 0.121 NS |
| GR | 5 | 20.82 | 0.000 ** | 5 | 1.72 | 0.000 ** | 1 | 8.45 | 0.005 ** | 25 | 1.24 | 0.206 NS | 5 | 8.47 | 0.000 ** | 25 | 1.24 | 0.201 NS | 5 | 0.74 | 0.601 NS |
| LR | 5 | 23.49 | 0.000 ** | 5 | 1.72 | 0.160 NS | 1 | 138.52 | 0.000 ** | 25 | 1.69 | 0.023 * | 5 | 15.93 | 0.000 ** | 25 | 1.64 | 0.031 * | 5 | 0.85 | 0.521 NS |
| NR | 5 | 29.23 | 0.000 ** | 5 | 24.21 | 0.000 ** | 1 | 19.24 | 0.000 ** | 25 | 1.77 | 0.015 * | 5 | 3.37 | 0.010 ** | 25 | 2.32 | 0.001 ** | 5 | 2.32 | 0.054 * |

Note: N, G and Y represent nitrogen, grazing and year, respectively. SR, GR, LR and NR represent species richness, grasses richness, legumes richness, forbs richness, respectively. NS = not significant, * = $p < 0.05$, ** = $p < 0.01$.

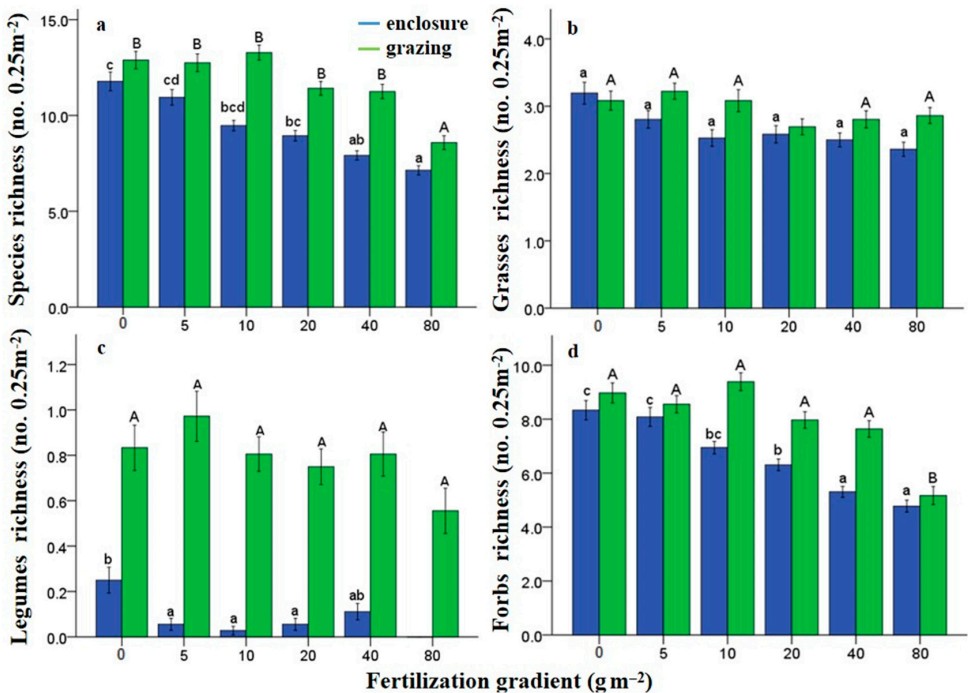

**Figure 3.** The response of species richness (**a**) and the richness of species across functional groups (grasses (**b**), legumes (**c**) and forbs (**d**)) to increasing levels of N in the enclosure and grazing grassland ecosystems; the green and blue columns represent the grazing and enclosure grassland ecosystems, respectively. Different lower-case and upper-case letters indicate significant differences between mean values within the columns for the enclosure grassland ecosystem and grazing grassland ecosystem, respectively, according to Tukey's tests at 5% probability.

A significantly positive relationship was recorded between the species richness and community stability in the enclosure and grazing grassland ecosystems (Figures 1 and 4a). Species richness along with community stability consistently declined during the course of the six-year experiment in the enclosure and grazing grassland ecosystems, and this decline was accelerated by N addition. These results indicate that the reduced community stability in our study was possibly associated with reduced species diversity. With the addition of N, a significantly negative relationship was noted between the richness and stability of grasses, while there was a positive relationship between the richness and stability of forbs (Figure A6a,c). However, no significant relationship was recorded between the richness and stability of legumes (Figure A6b).

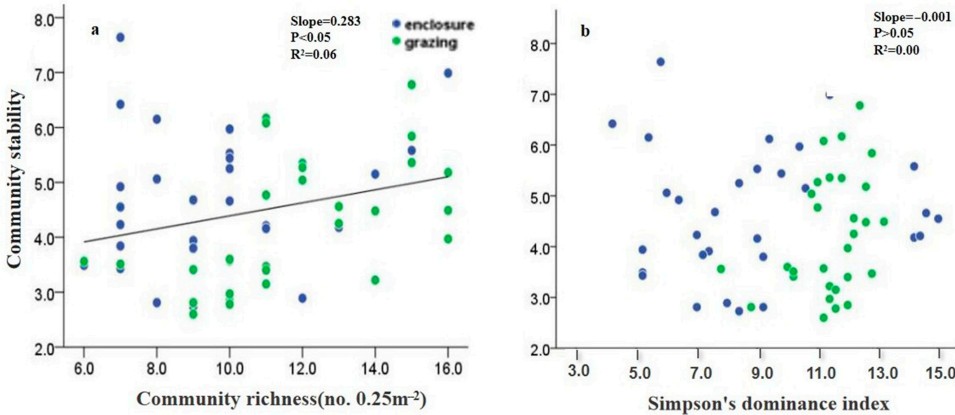

**Figure 4.** Linear relationship of species richness (**a**) and Simpson's dominance index (**b**) with community stability across six levels of N. The green and blue dots represent the grazing and enclosure grassland ecosystems, respectively.

### 3.3. Effects of N Fertilization and Grazing on Compensatory Dynamics

At all N levels, the community-wide synchrony of both species and functional groups was less than 1 in the enclosure and grazing grassland ecosystems (Figure 5). However, it was noted that the species synchrony and functional groups synchrony were greater in the enclosure grassland ecosystem than those of the grazing grassland ecosystem (Figure 5, suggesting a higher degree of compensatory dynamics among species/functional groups in the grazing ecosystem. However, N addition did not significantly improve the community-wide synchrony of either the species or the functional groups in the enclosure and grazing grassland ecosystems (Figures 1 and 5, Table A3). It was noted that grazing had significant effects on the community-wide synchrony of both species and functional groups (Table A3). However, no interactive effects of N versus grazing on the community-wide synchrony of both species and functional groups were recorded (Table A3). Overall, these results indicate that compensatory dynamics did not contribute to community stability under N addition and grazing.

### 3.4. Effects of N Fertilization and Grazing on Mean–Variance Scaling

There was a positive correlation between the log-transformed values of variance of species biomass and the log-transformed values of mean biomass at all levels of N addition treatments in the enclosure and grazing grassland ecosystems (Figure 6). However, N addition did not significantly affect the slopes (z, the scaling power) in the enclosure and grazing grassland ecosystems (Figures 1 and 6, Table A4). These results suggest that the mean–variance scaling was not the main mechanism underlying the reduction in community stability under the conditions of N addition and grazing.

### 3.5. Effects of N Fertilization and Grazing on Dominance

N addition and grazing resulted in significant changes in species dominance as indicated by Simpson's dominance index (Table A5). Simpson's dominance index decreased significantly with increasing the level of N in the enclosure and grazing grassland ecosystems (Figure 7). The results exhibited that there were significant interactive effects of grazing versus year and N addition versus year on the Simpson's dominance index (Table A5). However, there were no interactive effects of N addition versus grazing on the Simpson's dominance index (Table A5). In addition, there was no clear relationship between Simpson's dominance index and community stability (Figures 1 and 4b). These results suggested that species dominance had no roles in community stability in either the enclosure or grazing grassland ecosystems under N fertilization and grazing.

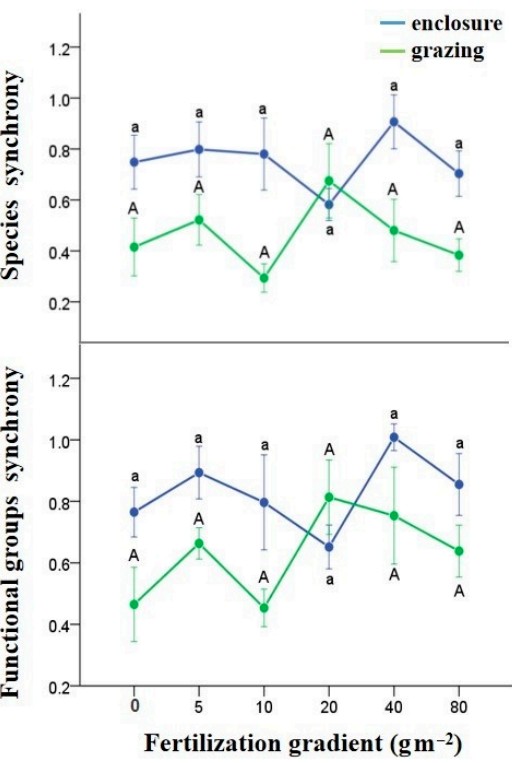

**Figure 5.** The response of species synchrony and functional groups synchrony to increasing levels of N in the enclosure and grazing grassland ecosystems. The green and blue lines represent the grazing and enclosure grassland ecosystems, respectively. Different lower-case and upper-case letters indicate significant differences between mean values within the data points for the enclosure grassland ecosystem and grazing grassland ecosystem, respectively, according to Tukey's tests at 5% probability.

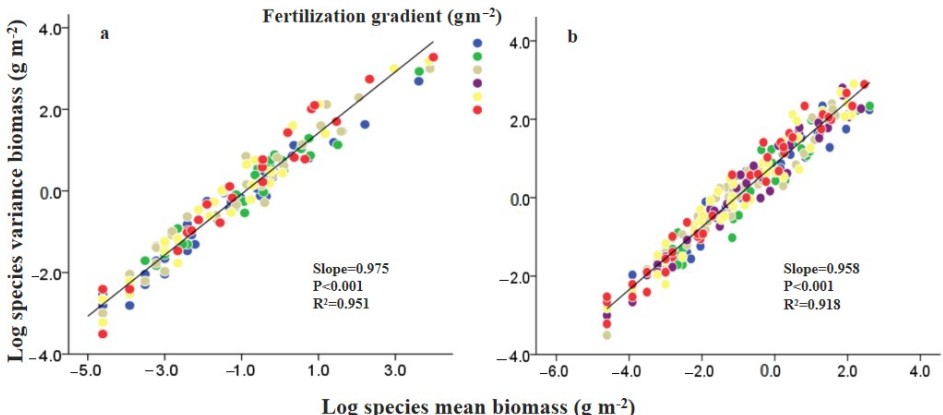

**Figure 6.** The relationship between the logarithms of variance and mean biomass across six levels of N fertilization in the enclosure (**a**) and grazing (**b**) grassland ecosystems, respectively. Different levels of N, namely, 0, 5, 10, 20, 40, 80 (g m$^{-2}$), are indicated by blue, green, purple, beige, yellow and red dots, respectively.

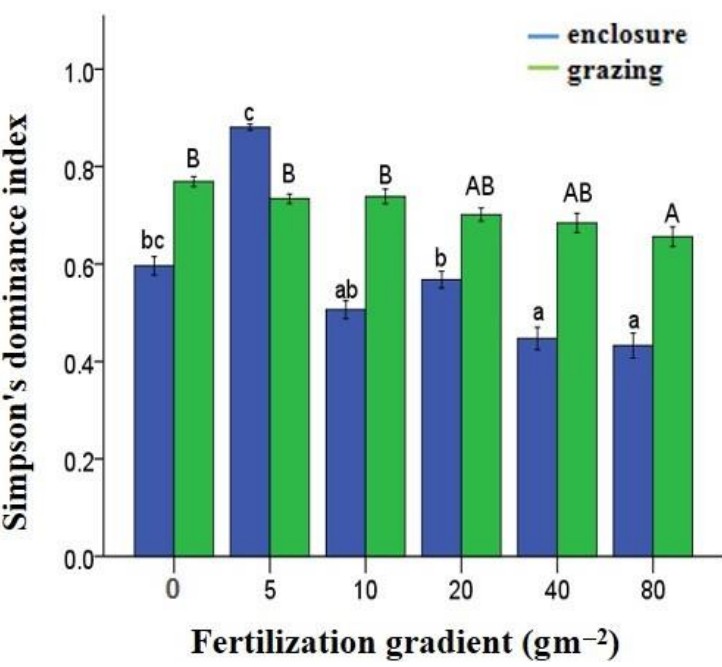

**Figure 7.** The response of community Simpson's dominance index to increasing levels of N in the enclosure and grazing grassland ecosystems. The green and blue columns represent the grazing and enclosure, respectively. Different lower-case and upper-case letters indicate significant differences between mean values within the columns for the enclosure grassland ecosystem and grazing grassland ecosystem, respectively, according to Tukey's tests at 5% probability.

## 4. Discussion

### 4.1. Effects of Fertilization and Grazing on Community Stability

Several studies have shown that fertilization and grazing can affect grassland ecosystem composition and stability [6,11,48]. Yang et al. [7] found that increasing the level of N increased the community stability on the Tibetan Plateau over 10 years. However, in our study, increasing the level of nitrogen reduced the community stability in the enclosure grassland and grazing grassland ecosystems due to the negative impacts of N addition on functional groups (Figures 1 and 2, Table A1). Factors such as species diversity and dominant vegetation can influence the response of community stability to perturbations [49]. Thus, the differences between the results from this research and those recorded by Yang et al. [7] could be due to the differences in species composition, species richness and dominant vegetation between both studies. Similar to our results, Wang et al. [12] found that community stability declined with N addition in an alpine meadow. Also, Song and Yu [1] reported a decline in community stability in an alpine meadow on the Tibetan Plateau as the result of N addition.

Our results showed that grazing had negative effects on the community stability and N addition enhanced the negative effects of grazing on the community stability in the grassland ecosystem. It appears that enclosure of grasslands is more conducive to community stability, supporting the national policy of banning grazing in grassland ecosystems in China [50]. Similarly, Ren et al. [51] found that the temporal community stability in the Eurasian steppe was reduced due to the adverse effects of grazing on compensatory dynamics. Also Salgado-Luarte et al. [8] found that grazing reduced the richness of species and altered the species composition. However, Beck et al. [52] found that moderate grazing promoted the community stability in California's largest serpentine grassland via limiting the infestation of exotic weeds and maintaining native plant communities. Also, Li et al. [53] found that rotational grazing increased community stability as grazing improved community productivity and maintained the compensatory growth of the plants on the Qinghai Tibetan plateau ecosystem [53]. The differential responses of community

stability to grazing recorded between our study and those by the Beck et al. [52] and Li et al. [53] could be due to differential grazing methods, climate regions, and vegetation types [49,52].

*4.2. Effects of Fertilization and Grazing on the Underlying Mechanisms Contributed to Community Stability*

Previous studies have indicated that perturbations affected community stability via several mechanisms, namely, species diversity, compensatory dynamics, mean–variance scaling and species dominance [6,53]. In our study, a positive relationship between species richness and community stability was noted. It was also noted that increasing the level of N significantly reduced species richness, thus resulting in a significant decline in the community stability in the enclosure and grazing grassland ecosystems (Figures 1 and 4a). This result is consistent with several other studies in which a positive correlation between species richness and community stability was noted, and the loss of community stability was found to be due to the negative impacts of perturbations on species diversity [17,28, 40,54–58]. Our results also indicated that species richness in the grazing ecosystem was higher than that in the enclosure ecosystem (Figure 3a), Niu et al. also found that grazing increased species richness as grazing promoted a reduction in competition for light among plants [59]. Similar results have been reported by Taddese et al. [60] and Anna et al. [61]. However, in other studies, species richness did not play a key role in the loss of community stability after fertilization [62,63].

The results of our study showed that in the grazing grassland ecosystem, the species richness reduction was accompanied by a decline in the richness of forbs, while no effects on the dominance of grasses and legumes were recorded following N addition. Tian et al. [64] noted that the addition of N resulted in reduced abundance of forbs while the abundance of grasses remained unchanged even with increasing the level of N. They suggested that differential responses recorded between forbs and grasses to N addition were attributed to the differential sensitivity of forbs versus grasses to soil acidification and mobilization of soil $Mn^{2+}$ induced by N. In the enclosure grassland ecosystem, N addition, however, resulted in a loss in the dominance of forbs and legumes, while it had no effects on the dominance of grasses. Midolo et al. [65] also noted that N addition induced significant losses in the abundance of legumes compared to other species. The primary mechanism underlying species richness losses as a result of N addition is enhanced light limitation [16]. The losses of legumes as a result of N addition could be due to increasing the abundance of tall species, which can limit ground-level light for short species [1,66]. In the present study, it was noted that while the addition of N resulted in a significant increase in the abundance of tall grass species such as *Elymus dahuricus,* there was a reduction in the abundance of short legume species such as *Medicago archiducis-nicolai* (Table A2). The greater abundance of legumes in the grazing grassland ecosystem compared to the enclosure grassland ecosystem implies that grazing could alleviate the negative impacts of the N-induced reduction in the richness of legumes, as suggested by Song and Yu [1].

Compensatory dynamics have been found to play a pivotal role in ecosystem stability after perturbations [1,54,62]. Compensatory dynamics mean that different species respond differently to environmental drivers, thus at the community level, the loss of one species will be compensated for by an increase of other species [5]. In our study, N addition did not significantly affect the synchrony of either species or functional groups in the enclosure and grazing grassland ecosystems, indicating that compensatory dynamics did not play a role in the community stability in either ecosystem (Figures 1 and 5). Similarly, Yang et al. [7] and Leps et al. [27] failed to find a role for species asynchrony in maintaining ecosystem stability after fertilization. In addition, it was noted that the interactive effects of N addition versus grazing on the species synchrony and functional groups synchrony were not significant, indicating compensatory dynamics had no role in community stability under conditions of N fertilization and grazing. The lack of interactive effects of N addition versus grazing on the synchrony of both species and functional groups may indicate that

the coexisting species in the studied ecosystems shared a common resource base and responded to both grazing and N addition in a similar way [19].

The mean–variance scaling is another mechanism that is known to maintain community stability. Perturbations can alter the slope of this mean–variance scaling relationship and result in variations in ecosystem stability [12,66]. For instance, Grman et al. [6] noted that an alteration in the stability of a grassland in Michigan, USA, was associated with a change in the slope of the mean–variance scaling relationship as a result of fertilization. However, our results showed that N addition, grazing and their interactions did not cause any alterations in the scaling power z; thus, the community stability in our case was not associated with the mean–variance scaling relationship mechanism (Figures 1 and 6). Similarly, Yang et al. [7] and Niu et al. [21] noted that the mean–variance scaling relationship mechanism was not associated with community stability in grassland ecosystems. According to Song and Yu [1], asynchronous species fluctuations in response to perturbations could be a reason for the lack of effects of mean–variance scaling on community stability in ecosystems.

Several studies have shown that increasing the abundance of dominant species can increase ecosystem stability [6,28]. However, our results showed that there was no correlation between Simpson's dominance index and community stability (Figures 1 and 4b), indicating that dominance was not the main mechanism driving community stability. Similarly, other studies noted that increasing the abundance of dominant species was not necessarily associated with greater stability in ecosystems [2,7,29].

Overall, our results showed that under N addition and grazing, there were differences between the grassland ecosystems on the Loess Plateau and those on the Inner Mongolia Plateau and Tibet Plateau in mechanisms underlying community stability [1,7,67]. These differences could be due to differences in the functional diversity of species on the Loess Plateau compared to the other ecosystems [20,62]. Another plausible reason is that in our study, we evaluated the effects of two anthropogenic changes (grazing and N addition) on community stability in contrast to many other studies in which the response of community stability was evaluated only under N addition [7,21].

## 5. Conclusions

Our results showed that increasing N addition reduced the community stability in the enclosure grassland and grazing grassland ecosystems. However, a long-term enclosure can alleviate the negative effects of fertilization on the grassland community stability on the Loess Plateau. In addition, at the mechanistic level, we found that compensatory dynamics, mean–variance scaling relationship and species dominance did not contribute to the grassland community stability on the Loess Plateau; in response to perturbations, however, the community stability was positively correlated with species diversity (Figure 1).

The results of this study provided a scientific basis for grassland management on the Loess Plateau. Work is in progress to (1) understand why species richness in the grazing grassland ecosystem is greater than in the enclosure grassland ecosystem when under perturbations, (2) understand what other factors can also contribute to the community stability in the enclosure and grazing grassland ecosystems on the Loess Plateau under anthropogenic changes.

**Author Contributions:** Conceptualization, investigation, data analysis, visualization, interpretation and writing the original draft, J.L.; conceptualization, investigation and data analysis, X.Y. (Xue Yang); interpretation, writing, reviewing and editing, H.G.; investigation, data analysis and investigation, X.Y. (Xinhui Yan), Y.F.; investigation, Q.G. and B.Z.; reviewing and editing, Z.W.; conceptualization, supervision, writing, reviewing and editing, W.L. All authors have read and agreed to the published version of the manuscript.

**Funding:** This research is funded by the Natural Science Foundation of Shaanxi Province of China (Grant No. 2020JM-162), the National Natural Science Foundation of China (Grant No. 41601586, 41671289), and the National Key Research and Development Program of China (Project No. 2016YFC0500700).

**Institutional Review Board Statement:** Not applicable.

**Informed Consent Statement:** Not applicable.

**Data Availability Statement:** The raw data required to reproduce these findings cannot be shared at this time as the data also forms part of an ongoing study.

**Acknowledgments:** We are grateful to the editor and anonymous reviewers for their constructive comments and suggestions.

**Conflicts of Interest:** The authors declare no conflict of interest.

## Appendix A

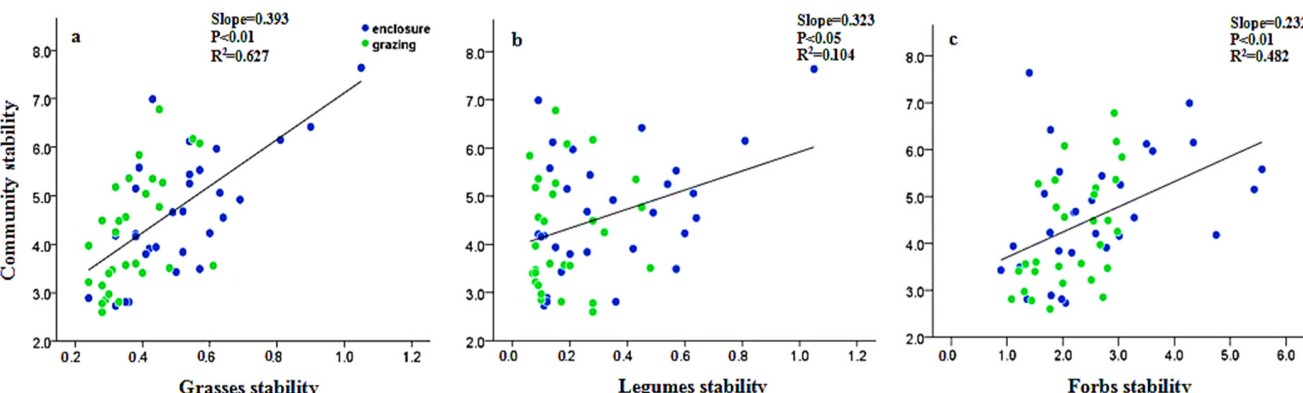

**Figure A1.** The relationship between community stability and the stability of species across functional groups (grasses (**a**), legumes (**b**) and forbs (**c**)). The green and blue dots represent the grazing and enclosure grassland ecosystems, respectively.

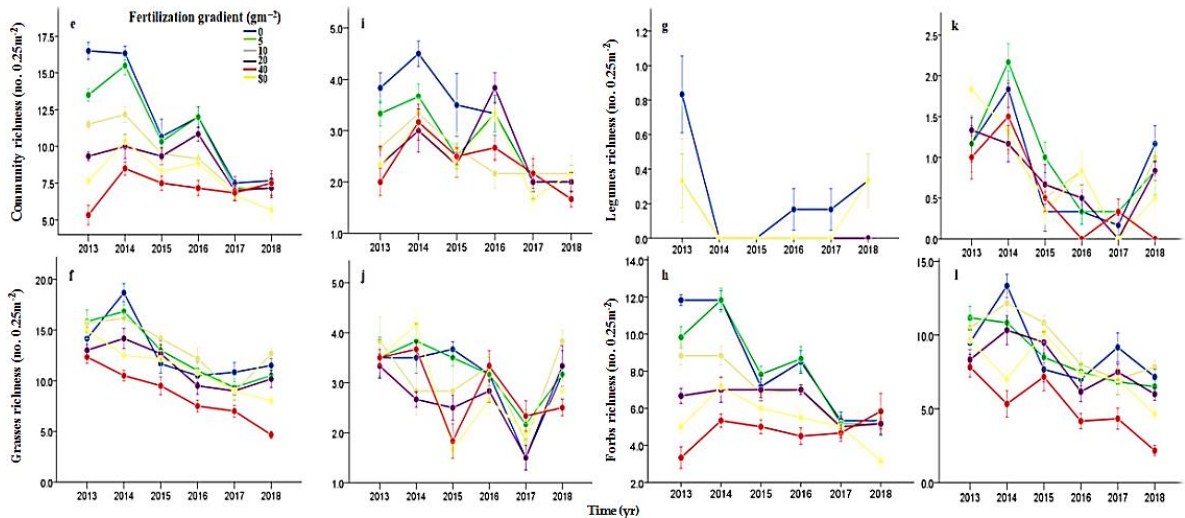

**Figure A2.** The temporal tends of community richness (**e**,**i**) and the richness of species across functional groups (grasses (**f**,**j**), legumes (**g**,**k**) and forbs (**h**,**l**)) with increasing the level of N in the enclosure (**e**–**h**) and grazing (**i**–**l**) grassland ecosystems, respectively, during the course of the six-year experiment. Different levels of N, namely, 0, 5, 10, 20, 40, 80 (g m$^{-2}$), are indicated by blue, green, gray, purple, red and yellow lines, respectively.

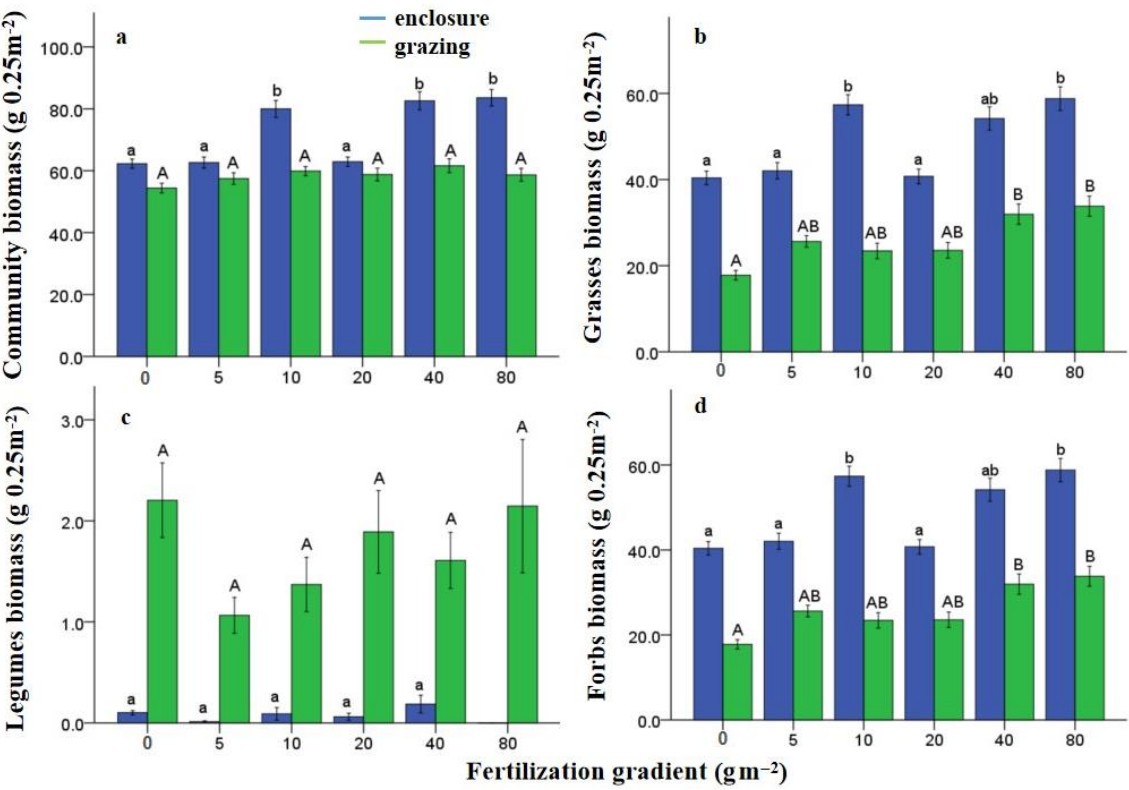

**Figure A3.** The response of community biomass (**a**) and the biomass of species across functional groups (grasses (**b**), legumes (**c**) and forbs (**d**)) to increasing levels of N in the enclosure and grazing grassland ecosystems. The green and blue columns represent the grazing and enclosure grassland ecosystems, respectively. Different lower-case and upper-case letters indicate significant differences between mean values within the columns for the enclosure grassland ecosystem and grazing grassland ecosystem, respectively, according to Tukey's tests at 5% probability.

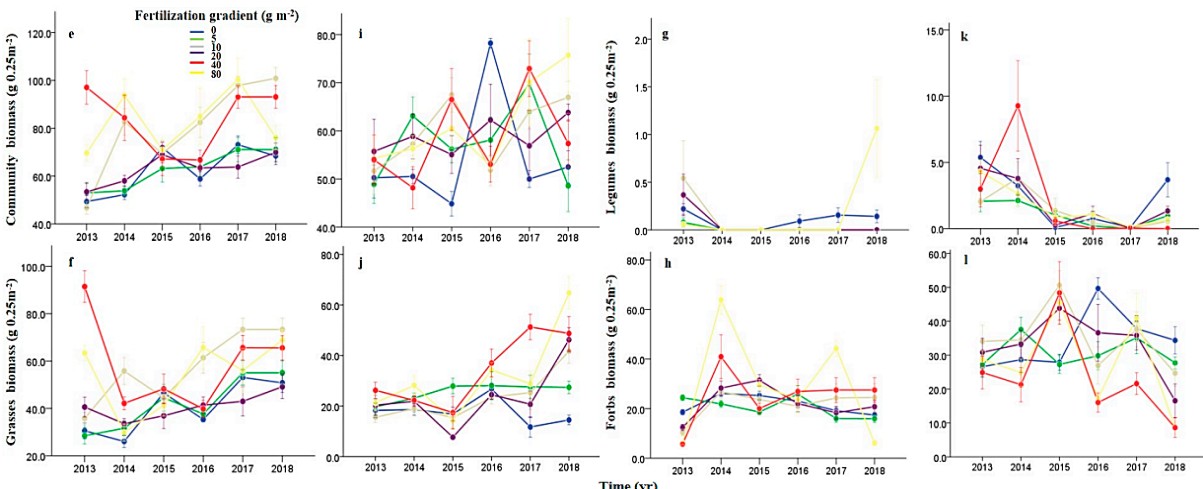

**Figure A4.** The temporal trends of community biomass (**e,i**) and the biomass of species across functional groups (grasses (**f,j**), legumes (**g,k**) and forbs (**h,l**)) with increasing the level of N in the enclosure (**e–h**) and grazing (**i–l**) grassland ecosystems, during the course of the six-year experiment. Different levels of N, namely, 0, 5, 10, 20, 40, 80 (g m$^{-2}$), are indicated by blue, green, beige, purple, red and yellow lines, respectively.

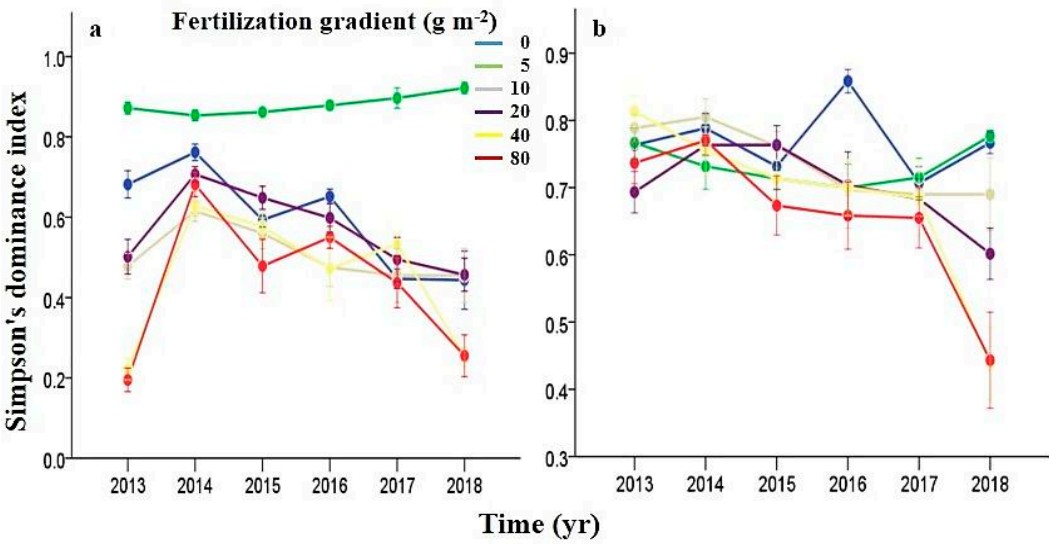

**Figure A5.** The general trends of Simpson's dominance index with increasing the level of N in the enclosure (**a**) and grazing (**b**) grassland ecosystems, during the course of the six-year experiment. Different levels of N, namely, 0, 5, 10, 20, 40, 80 (g m$^{-2}$), are indicated by blue, green, beige, purple, red and yellow lines, respectively.

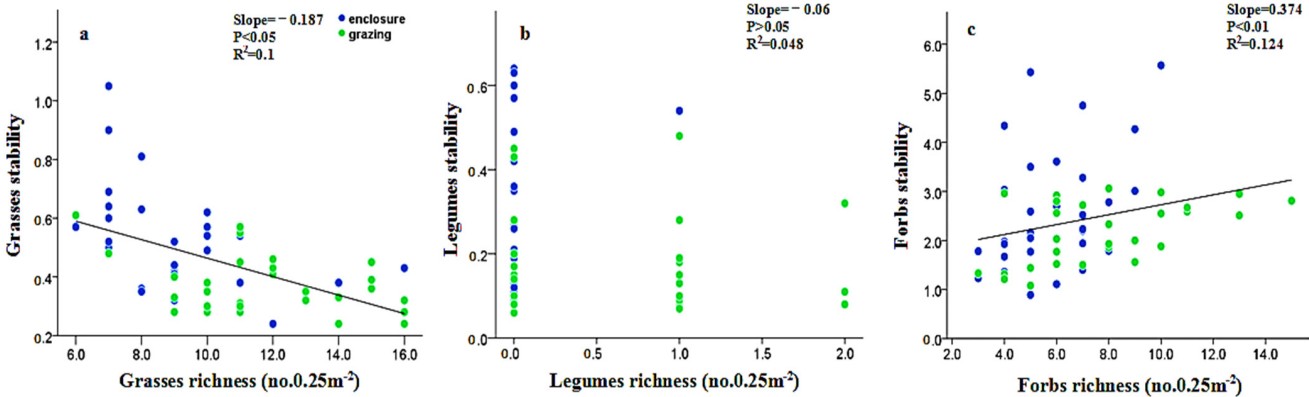

**Figure A6.** The relationship between the stability and richness of species across functional groups (grasses (**a**), legumes (**b**) and forbs (**c**)). The green and blue dots represent the grazing and enclosure grassland ecosystems, respectively.

## Appendix B

**Table A1.** Results of two-way repeated-measure ANOVAs of effects of nitrogen, grazing and their interaction on community stability.

| | N | | | G | | | N*G | | |
|---|---|---|---|---|---|---|---|---|---|
| | df | F | p | df | F | p | df | F | p |
| COS | 5 | 4.25 | 0.007 ** | 1 | 3.41 | 0.018 * | 5 | 2.71 | 0.031 * |
| GRS | 5 | 3.48 | 0.017 * | 1 | 0.90 | 0.495 NS | 5 | 2.22 | 0.068 NS |
| NFS | 5 | 8.44 | 0.000 ** | 1 | 8.32 | 0.000 ** | 5 | 1.04 | 0.403 NS |
| LES | 5 | 0.57 | 0.692 NS | 1 | 0.69 | 0.640 NS | 4 | 0.27 | 0.893 NS |

Note: N, G represent nitrogen and grazing, COS, GRS, NFS and LES represent community stability, grasses stability, forbs stability and legumes stability, respectively. NS = not significant, * = $p < 0.05$, ** = $p < 0.01$.

**Table A2.** The effect of N and grazing on mean population stability in the enclosure and grazing grassland ecosystems. Blank cells indicate the species was not present throughout the experimental period. F-values from two-way ANOVA show the effect of nitrogen and grazing on mean population stability. NS = not significant, * = $p < 0.05$, ** = $p < 0.01$.

| Functional Groups | Species Name | Mean Population Stability | | | | | | | | | | | | N | G | N*G |
| | | Enclosure | | | | | | Grazing | | | | | | | | |
| | | 0 | 5 | 10 | 20 | 40 | 80 | 0 | 5 | 10 | 20 | 40 | 80 | | | |
| Grasses | *Stipa przewalskyi* | 3.72 | 2.50 | 2.94 | 2.67 | 2.07 | 2.71 | 1.01 | 1.01 | 1.06 | 1.01 | 1.01 | 1.19 | 1.18 NS | 166.60 ** | 1.24 NS |
| | *Carex aridula* | 0.96 | 0.87 | 0.96 | 0.81 | 0.63 | 0.54 | 1.37 | 2.40 | 2.41 | 1.47 | 1.01 | 1.01 | 3.63 ** | 32.06 ** | 0.49 NS |
| | *Leymus secalinus* | 0.56 | 0.65 | 0.65 | 0.63 | 0.45 | 0.72 | 1.01 | 1.25 | 1.11 | 1.09 | 1.05 | 1.01 | 0.45 NS | 47.72 ** | 0.35 NS |
| | *Elymus dahuricus* | 0.79 | 0.48 | 0.64 | 0.64 | 0.74 | 0.40 | 1.01 | 1.01 | 1.01 | 1.01 | 1.01 | 1.01 | 4.33 ** | 0.45 NS | 0.51 NS |
| | *Poa annua* | 0.67 | 0.58 | 0.39 | 0.36 | 0.35 | 0.31 | 1.74 | 3.06 | 1.01 | 1.01 | 1.02 | 1.02 | 4.82 ** | 57.68 ** | 0.58 NS |
| | *Stipa grandis* | 0.65 | 0.44 | 0.46 | 0.46 | 0.39 | 0.38 | 2.01 | 1.91 | 1.02 | 1.02 | 1.01 | 1.01 | 2.87* | 51.56 ** | 0.91 NS |
| | *Agropyron cristatum* | 0.37 | 0.35 | 0.30 | 0.36 | 0.30 | 0.31 | | | | | | | 1.60 NS | | |
| Legumes | *Medicago archiducis-nicolai* | 0.45 | 0.35 | 0.31 | 0.36 | 0.42 | | 1.01 | 1.01 | 1.02 | 1.02 | 1.37 | | 0.29 NS | 25.14 ** | 0.11 NS |
| | *Astragalus tataricus* | 0.41 | 0.31 | | | 0.43 | | 1.01 | 1.05 | 1.02 | 1.13 | 1.02 | 1.01 | 1.26 NS | 113.41 ** | 4.36* |
| | *Oxytropis ochrantha* var. *ochrantha* | 0.28 | | | | 0.31 | | 1.01 | | | 1.58 | 1.01 | 1.01 | 0.38 NS | 5.54* | 4.54* |
| Forbs | *Heteropappus altaicus* | 0.45 | 0.45 | 0.53 | 0.62 | 0.39 | 0.31 | 1.01 | 2.36 | 1.88 | 3.31 | 1.18 | 1.13 | 2.81* | 71.41 ** | 0.40 NS |
| | *Artemisia sacrorum* | 1.38 | 1.19 | 0.87 | 1.47 | 1.23 | 0.77 | 1.01 | 1.71 | 1.23 | 1.01 | 1.55 | | 0.41 NS | 0.46 NS | 0.31 NS |
| | *Thymus mongolicus* | 0.43 | 0.59 | 0.41 | 0.43 | 0.39 | 0.31 | 4.16 | 2.03 | 1.39 | 2.94 | 3.61 | 1.44 | 1.07 NS | 87.10 ** | 0.65 NS |
| | *Crepis crocea* | 0.37 | 0.37 | 0.45 | 0.36 | 0.31 | | | | | | | | 0.24 NS | | |
| | *Trigonotis amblyosepala* | 0.42 | 0.31 | | 0.31 | 0.30 | | | | | | | | 0.23 NS | | |
| | *Potentilla bifurca* | 0.65 | 0.76 | 0.68 | 0.68 | 0.66 | 1.04 | | | 1.01 | | 1.01 | 1.01 | 1.60 NS | 0.43 NS | 2.34 NS |
| | *Dendranthema lavandulifolium* | 1.04 | 1.09 | 0.77 | 0.59 | 0.72 | 0.65 | 1.64 | 1.09 | 1.011 | 1.01 | 1.92 | 5.21 | 1.13 NS | 14.61 NS | 1.37 NS |
| | *Leontopodium leontopodioides* | 0.54 | 0.64 | 0.45 | 0.63 | 0.34 | 0.43 | 1.09 | 3.97 | 1.04 | 1.13 | 1.02 | 1.01 | 11.45 ** | 145.05 ** | 5.11 ** |
| | *Viola dissecta* | 0.57 | 0.61 | 0.71 | 0.81 | 0.62 | 0.55 | | 1.10 | | | 1.02 | | 1.29 NS | 12.55 ** | 0.41 NS |
| | *Galium verum* | 1.43 | 1.67 | 1.39 | 1.08 | 0.95 | 0.91 | | | | | | | 4.51 ** | | |
| | *Saussurea alata* | 0.63 | 0.58 | 0.65 | 0.51 | 0.33 | 0.54 | | | | | | | 1.26 NS | | |
| | *Dracocephalum heterophyllum* | 0.31 | | | 0.37 | | | 1.01 | 5.87 | 1.01 | 1.01 | | | 0.63 NS | 84.10 ** | 0.02 ** |
| | *Scutellaria scordifolia* | 0.70 | 0.70 | 0.75 | 0.52 | 0.60 | 0.50 | | | | | | | 1.78 NS | | |
| | *Artemisia gansuensis* | 0.31 | 0.31 | 0.42 | 0.30 | 0.31 | | 1.33 | 1.64 | 1.05 | 2.85 | 1.12 | 1.08 | 0.71 NS | 63.86 ** | 1.14 NS |
| | *Viola prionantha* | 0.41 | 0.35 | 0.41 | 0.31 | | 0.37 | | | | | | | 0.18 NS | | |
| | *Thalictrum petaloideum* | 0.41 | 0.42 | 0.41 | 0.35 | 0.30 | 0.35 | | | | | | | 1.01 NS | | |
| | *Polygala tenuifolia* | 0.31 | 0.46 | 0.31 | 0.37 | | | 1.01 | 1.01 | 1.01 | 1.32 | 1.01 | | 0.73 NS | 77.79 ** | 0.51 NS |
| | *Potentilla acaulis* | 0.35 | 0.34 | 0.31 | 0.31 | 0.37 | | 1.01 | 1.17 | 1.01 | 1.01 | | | 1.01 NS | 68.4 ** | 0.66 NS |
| | *Adenophora stenanthina* | 0.59 | 0.71 | 0.43 | 0.53 | 0.42 | 0.41 | | | | | | | 2.47* | | |
| | *Androsace mariae* | 0.31 | | 0.31 | 0.30 | | | 1.01 | 1.02 | 1.01 | 1.01 | 1.01 | | 0.01 NS | 1.24 NS | 0.02 NS |
| | *Allium ramosum* | 0.31 | 0.31 | 0.31 | 0.35 | 0.31 | 0.30 | | | | | | | 0.89 NS | | |
| | *Salsola collina* | 0.76 | 0.93 | 0.66 | 0.57 | 0.63 | 0.46 | | | | | | | 0.50 NS | | |
| | *Adenophora paniculata* | 0.37 | 0.34 | 0.45 | 0.51 | 0.32 | 0.33 | | 1.01 | 1.01 | | | | 1.75 NS | 49.40 NS | 0.77 NS |
| | *Torularia humilis* | 0.34 | 0.41 | 0.33 | 0.30 | 0.34 | 0.31 | | | | | | | 2.05 NS | | |
| | *Euphorbia fischeriana* | | | | | | | 1.01 | 1.02 | 1.01 | 1.01 | 1.01 | 1.01 | | 0.38 NS | |

Note: N and G represent nitrogen and grazing, respectively.

**Table A3.** The results of two-way repeated-measure ANOVAs of effects of nitrogen, grazing and their interaction on species synchrony and functional groups synchrony.

|  | N | | | G | | | N*G | | |
|---|---|---|---|---|---|---|---|---|---|
|  | df | F | *p* | df | F | *p* | df | F | *p* |
| SSY | 5 | 0.46 | 0.807 [NS] | 1 | 13.81 | 0.000 ** | 5 | 0.50 | 0.774 [NS] |
| FSY | 5 | 1.05 | 0.410 [NS] | 1 | 6.94 | 0.011 * | 5 | 0.55 | 0.741 [NS] |

Note: N and G represent nitrogen and grazing, respectively. SSY and FSY represent species synchrony and functional groups synchrony, respectively. NS = not significant, * = $p < 0.05$, ** = $p < 0.01$.

**Table A4.** The results of two-way repeated-measure ANOVAs of effects of nitrogen, grazing and their interaction on the scaling power z.

|  | N | | | G | | | N*G | | |
|---|---|---|---|---|---|---|---|---|---|
|  | df | F | *p* | df | F | *p* | df | F | *p* |
| Z | 5 | 0.60 | 0.702 [NS] | 1 | 0.49 | 0.485 [NS] | 5 | 0.29 | 0.916 [NS] |

Note: N, G represent fertilization and grazing, NS = not significant, * = $p < 0.05$.

**Table A5.** The results of two-way repeated-measure ANOVAs of effects of year, nitrogen, grazing and their interaction on Simpson's dominance index.

|  | Y | | | N | | | G | | | Y*N | | | Y*G | | | Y*G*N | | | N*G | | |
|---|---|---|---|---|---|---|---|---|---|---|---|---|---|---|---|---|---|---|---|---|---|
|  | df | F | *p* | df | F | *p* | df | F | *p* | df | F | *p* | df | F | *p* | df | F | *p* | df | F | *p* |
| DI | 5 | 31.86 | 0.000 ** | 5 | 10.25 | 0.000 ** | 1 | 86.26 | 0.000 ** | 25 | 2.25 | 0.001 ** | 5 | 4.51 | 0.002 ** | 25 | 2.21 | 0.001 ** | 5 | 1.62 | 0.168 [NS] |

Note: N, G and Y represent nitrogen, grazing and year, respectively. DI represents community Simpson's dominance index, NS = not significant, * = $p < 0.05$, ** = $p < 0.01$

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
