# Peer review of "Long-Term Enclosure Can Benefit Grassland Community Stability on the Loess Plateau of China"

_sustainability, doi:10.3390/su13010213_

Round 1

Reviewer 1 Report

This was an interesting, well-written, and thorough study, adeptly examining the interactions of grazing and nitrogen inputs on a grassland ecosystem of global significance.

While the framing of these ecological dynamics under two types of anthropogenic disturbance was focused on compensatory dynamics, I feel the piece would also benefit from examining the literature on State-and-Transition Models (STMs). Non-equilibrium systems are rarely stable over time even without disturbance in arid or semi-arid grasslands, and STMs are a useful way to examine changes in community composition with both anthropogenic or natural disturbances. With or without STMs being considered in the paper, I think discussion of non-equilibrium dynamics in more detail would improve the piece.

The only portion of the paper I found somewhat lacking was the discussion. For example, the paragraph beginning at 313 mentioned many contradictory studies on the effects of grazing without going into any detail as to why different regions had different responses, and how this study fits into those overall discussions. And the finding in this study that grazing led to greater species richness under perturbation was quite central, but not particularly discussed. To me, this is really the point of the piece. Does exclosure work for preserving ecosystem function? The authors mention on line 243 that species richness and stability "consistently declined" during the experiment, both within exclosures and grazed pastures. Moreover, exclosures have been shown to have negative impacts on pastoralists and ranchers (culturally and economically), hence the focus on community-based natural resource management in India, the United States, Mongolia, and other places with a history of grazing. In summary, I feel the authors should focus more on fleshing out how their results compare to other grazed vs exclosure studies, and why their results align with some studies and contradict others. Line 43 touches on this dichotomy, but the discussion didn't seem to return to the notion with the depth it deserves.

Below, please find my specific minor comments:
20: suggest splitting into two sentences after "Loess"
23: What unit? Presumably grams, but should be stated
35: Define "community stability" and state why it should matter. Aren't many grassland ecosystems non-equilibrium systems with immense natural variability in plant communities over time?
67: There is an opportunity here to discuss past research on grazing exclusion. Many studies find it has an adverse effect on conservation goals, and that should be mentioned in the context of China's national policies of 'Returning Grazing Lands to Grasslands.' Additionally, it would be useful to know the historic land management of this region, especially with regards to grazing (authors mention 38 years of grazing, but I mean historic in the longer sense).
103: is this a common stocking rate in the area, or higher or lower?
178: delete "the" in "the level"
186: Figure may benefit from defining "dominance" as Simpson's dominance index
252: May be worth noting the relationships are weak in spite of significance
317: There are many studies indicating grazing did not adversely impact stability in the Eurasian steppe and many arid shrublands as well. Why this disparity?
349: Conversely, Houlahan et al (2007) found that compensatory dynamics mattered far less than precipitation in the ecosystems they studied.

Author Response

List of responses to the comments- Sustainability-1047272

Dear Editor,

We appreciate the opportunity to resubmit our manuscript “Long-term enclosure can alleviate the effects of fertilization on grassland community stability on the Loess Plateau of China” (Sustainability-1047272). The majority of comments by the Editor and reviewers have been addressed, with corresponding changes made directly to the manuscript where appropriate. Accompanying this letter, please find a revised version of our manuscript. We used the Track Changes feature in MS Word in places where the reviewers asked for modifications or additional information. Please note that the line number of the revised version of our manuscript is different from the original file submitted in the beginning due to additional text made with these changes. The new line number has been used where appropriate.Please address correspondence concerning the manuscript to me at liwei2013@nwsuaf.edu.cn. Thank you for your consideration of this manuscript.

Yours sincerely,

Wei Li

In answer to Reviewer #1

  1. While the framing of these ecological dynamics under two types of anthropogenic disturbance was focused on compensatory dynamics, I feel the piece would also benefit from examining the literature on State-and-Transition Models (STMs). Non-equilibrium systems are rarely stable over time even without disturbance in arid or semi-arid grasslands, and STMs are a useful way to examine changes in community composition with both anthropogenic or natural disturbances. With or without STMs being considered in the paper, I think discussion of non-equilibrium dynamics in more detail would improve the piece.

Reply: We want to thank this reviewer for the suggestions. In our study, we used a structural equation model (SEM) rather than State-and-Transition Models (STMs) because SEMs are discrete. Also, SEMs are more general than STMs since SEMs can accommodate both concurrent effects and memory effects. Therefore, SEMs are deemed as a powerful and flexible mathematic language for describing complex dynamic network systems. In the non-equilibrium systems, the primary drivers of vegetation and livestock dynamics are stochastic factors such as rainfall which result in highly variable and unpredictable primary production. Hence, non-equilibrium models are not applicable to the results of our study as we did not evaluate such the impact of such factors on grassland community stability on the Loess Plateau of China.

  1. The only portion of the paper I found somewhat lacking was the discussion. For example, the paragraph beginning at 313 mentioned many contradictory studies on the effects of grazing without going into any detail as to why different regions had different responses, and how this study fits into those overall discussions.

Reply: Thanks for your suggestions. We have discussed the differences between our results and to others and possible reasons for that (Lines 319-332).

  1. And the finding in this study that grazing led to greater species richness under perturbation was quite central, but not particularly discussed

Reply: Thanks for your suggestions. We have further discussed the species richness under perturbation (Lines 341-344).

  1. To me, this is really the point of the piece. Does exclosure work for preserving ecosystem function? The authors mention on line 243 that species richness and stability "consistently declined" during the experiment, both within exclosures and grazed pastures.

Reply: Thanks for your question. As discussed in Lines 353-363, the results of our study showed that in the enclosure grassland ecosystem, nitrogen addition resulted in a loss in the dominance of legumes while it had no effects on grasses. So, nitrogen addition resulted in a decline in legumes thus, a reduction in the species richness in the enclosure grassland ecosystem. Grazing, however alleviate the negative effects of nitrogen on legumes thus, a greater abundance of legumes in the grazing grassland ecosystem was recorded. This implies that grazing (not enclosure) could preserve the ecosystem function under nitrogen application in our study. In other words, grazing can alleviate the negative effects of nitrogen on grassland ecosystem.

  1. Moreover, exclosures have been shown to have negative impacts on pastoralists and ranchers (culturally and economically), hence the focus on community-based natural resource management in India, the United States, Mongolia, and other places with a history of grazing. In summary, I feel the authors should focus more on fleshing out how their results compare to other grazed vs exclosure studies, and why their results align with some studies and contradict others.

Reply: We thanks for your valuable suggestion. We have further discuss this by comparing our results to those of grazed vs exclosure studies in Lines 321-332 and 341-344.

  1. Line 43 touches on this dichotomy, but the discussion didn't seem to return to the notion with the depth it deserves.

Reply: Thanks for your suggestions. In lines 356-360, we mentioned that nitrogen addition results in the loss of legumes but promoted grasses. Then we mentioned that grazing however, alleviated this negative impacts of nitrogen on legumes.

  1. 20: suggest splitting into two sentences after "Loess"

Reply: Thanks for your suggestions. We have revised it.

  1. 23: What unit? Presumably grams, but should be stated

Reply: Thanks for your suggestions. We have modified it.

  1. 35: Define "community stability" and state why it should matter. Aren't many grassland ecosystems non-equilibrium systems with immense natural variability in plant communities over time?

Reply: Thanks for your valuable suggestion. We have define the term in Lines 33-38. It is true that many grassland ecosystems (non-equilibrium systems) have immense natural variability in plant communities over time, however, our objective was to assess how the stability of the grassland ecosystem changed when facing anthropogenic perturbations (fertilization and grazing) over a 6-year period from 2013 to 2018. As you mentioned, plant communities are continually undergoing change; however, we lack a clear understanding of how much change is typical and whether environmental characteristics affect rates of change. Further, it is unknown whether background rates of change relate to the responsiveness of a community to perturbations, and these are the contents of our further study.

  1. 67: There is an opportunity here to discuss past research on grazing exclusion. Many studies find it has an adverse effect on conservation goals, and that should be mentioned in the context of China's national policies of 'Returning Grazing Lands to Grasslands. Additionally, it would be useful to know the historic land management of this region, especially with regards to grazing (authors mention 38 years of grazing, but I mean historic in the longer sense).

Reply: Thanks for your valuable suggestion. The past research on grazing exclusion has been discussed in Lines 69-71 and 75-78. Historically, this region belongs to the typical agricultural and pastoral transition area.

  1. 103: is this a common stocking rate in the area, or higher or lower?

Reply: Thanks for your question. Three sheep ha-2 is a common stocking rate in the area.

  1. 178: delete "the" in "the level"

Reply: Thanks for your suggestions. We have revised it.

  1. 186: Figure may benefit from defining "dominance" as Simpson's dominance index

Reply: Thanks for your suggestions. We have revised it.

  1. 252: May be worth noting the relationships are weak in spite ofsignificance

Reply: Thanks for your question. The relationships may be weak but it can still significantly show the relationship between species richness and community stability in the enclosure and grazing grassland ecosystems. In addition, the ecosystem itself is a complex and dynamic system. In further study, we will combine more factors, such soil microorganism, to further clarify the internal mechanism.

  1. 317: There are many studies indicating grazing did not adversely impact stability in the Eurasian steppe and many arid shrublands as well. Why this disparity?

Reply: Thanks for your question. We have discussed the differences between our results and to others and possible reasons for that (Lines 321-332).

  1. 349: Conversely, Houlahan et al (2007) found that compensatory dynamics mattered far less than precipitation in the ecosystems they studied.

Reply: Thanks for your question and suggestion. It is because anthropogenic factors (grazing and fertilization) have greater impacts on compensatory dynamics among species than natural factors.

Reviewer 2 Report

The manuscript entitled „Long-term enclosure can alleviate the effects of fertilization on grassland community stability on the Loess Plateau of China”  by Jing Liu et al. is an interesting study aiming to assess the influence of nitrogen fertilization and grazing on the community stability in a long-term enclosure and grazing grassland ecosystems on the Loess Plateau of China. I found this manuscript as a well written article with interesting and valuable results in terms of their applications.

I have only few recommendations for the authors:

  1. I recommend that authors should consider rephrase the title since its present formulation looks like a research hypotheses and is quite similar with the major conclusion of the research.
  2. Line 18-20: The phrase is too long and difficult to understand. I recommend you to split it in 2 different phrases.

Suggestion: „We conducted a six-year field experiment to assess the influence of nitrogen (N) fertilization and grazing on the community stability in a long-term enclosure and grazing grassland ecosystems on the Loess Plateau. The structural equation modelling method was used to evaluate how fertilization and grazing altered community stability.”

  1. Line 26: I recommend you to delete the first part of the phrase since you already presented the same idea regarding the impact of N addition on comunity stability in line 21-22 (see below):

Line 21-22: Our results indicated that the community stability decreased in the enclosure and grazing grassland ecosystems with the addition of N.

Line 26: Overall, the addition of N impacted the community stability in the enclosure and grazing grassland ecosystems [...].

  1. You mentioned in Line 106 that the fertilizer was applied during a rain event to avoid the need for watering. What do you mean by „during a rain event”? Usually, as you already know fertilizer should be applied before/after approximately 2 hours a rain in order to avoid nutrient lavation.
  2. Please check the manuscript for minor spell checks and editing errors:

Line 47: I recommend you to use the word work or functioning instead of function in the following sentence: „predict how ecosystems function in response to perturbations”. Ecosystem function is the capacity of natural processes and components to provide goods and services that satisfy human needs, either directly or indirectly (de Groot et al 2002) but, as I understood, your phrase refers to how the ecosystems works or is functioning.

Line 61: Replace „by Taylor [23]” with „by Taylor et al [23]”.

Line 167-168: I think the text between parenthesis is subscript. Is it ok like that?

Line 194: the definition/signification of (a) is missing form Figure’s 2 title

Line 329: missing reference no. 53– you cited „perturbations on species diversity [17, 28, 46, 51, 52, 54].”

Line 339: You cited Midolo et al [20] but the reference no. 20 in References section is Wildova et al

Line 348: „Song et al” should be cited „Song and Yu”

Line 539: Reference [37] - Please edit the species name with italic (Pinus sylvestris)

Line 551: Please check if these are authors last name: „Emily, G.; Jennifer, A.L.; Donald, R.S. Mechanisms contributing to stability in ecosystem function depend on the environmental context. Ecology Letters. 2010. 13, 1400-1410.” – they look like first names for me.

Author Response

List of responses to the comments- Sustainability-1047272

Dear Editor,

We appreciate the opportunity to resubmit our manuscript “Long-term enclosure can alleviate the effects of fertilization on grassland community stability on the Loess Plateau of China” (Sustainability-1047272). The majority of comments by the Editor and reviewers have been addressed, with corresponding changes made directly to the manuscript where appropriate. Accompanying this letter, please find a revised version of our manuscript. We used the Track Changes feature in MS Word in places where the reviewers asked for modifications or additional information. Please note that the line number of the revised version of our manuscript is different from the original file submitted in the beginning due to additional text made with these changes. The new line number has been used where appropriate.Please address correspondence concerning the manuscript to me at liwei2013@nwsuaf.edu.cn. Thank you for your consideration of this manuscript.

Yours sincerely,

Wei Li

In answer to Reviewer #2 

  1. I recommend that authors should consider rephrase the title since its present formulation looks like a research hypotheses and is quite similar with the major conclusion of the research

Reply: Thanks for your suggestions. In this revision edition, we have changed the title to “Long-term enclosure can benefit grassland community stability on the Loess Plateau of China”

2.Line 18-20: The phrase is too long and difficult to understand. I recommend you to split it in 2 different phrases.

Suggestion: „We conducted a six-year field experiment to assess the influence of nitrogen (N) fertilization and grazing on the community stability in a long-term enclosure and grazing grassland ecosystems on the Loess Plateau. The structural equation modelling method was used to evaluate how fertilization and grazing altered community stability.”

Reply: Thanks for your suggestions.The sentence has been re-phrased as suggested by this reviewer.

3.Line 26: I recommend you to delete the first part of the phrase since you already presented the same idea regarding the impact of N addition on comunity stability in line 21-22 (see below):

Line 21-22: Our results indicated that the community stability decreased in the enclosure and grazing grassland ecosystems with the addition of N.

Line 26: Overall, the addition of N impacted the community stability in the enclosure and grazing grassland ecosystems [...].

Reply: Thanks for your suggestions.The sentence has been re-phrased as suggested by this reviewer.

4.You mentioned in Line 106 that the fertilizer was applied during a rain event to avoid the need fo watering. What do you mean by „during a rain event”? Usually, as you already know fertilizer should be applied before/after approximately 2 hours a rain in order to avoid nutrient lavation.

Reply:Thanks for your suggestions. We are sorry for the typo. This has been corrected.

Please check the manuscript for minor spell checks and editing errors:

Line 47: I recommend you to use the word work or functioning instead of function in the following sentence: „predict how ecosystems function in response to perturbations”. Ecosystem function is the capacity of natural processes and components to provide goods and services that satisfy human needs, either directly or indirectly (de Groot et al 2002) but, as I understood, your phrase refers to how the ecosystems works or is functioning.

Reply: Thanks for your suggestions.The sentence has been re-phrased as suggested by this reviewer.

  1. Line 61: Replace „by Taylor [23]” with „by Taylor et al [23]”.

Reply: Thanks for your suggestions.This has been corrected.

Line 167-168: I think the text between parenthesis is subscript. Is it ok like that?

Reply:Thanks for your suggestions. This has been corrected.

  1. Line 194: the definition/signification of (a) is missing form Figure’s 2 title

Reply: Thanks for your suggestions.This has been corrected.

  1. missing reference no. 53– you cited „perturbations on species diversity [17, 28, 46, 51, 52, 54].”

Reply:Thanks for your suggestions. This has been corrected.

Line 339: You cited Midolo et al [20] but the reference no. 20 in References section is Wildova et al

Reply: Thanks for your suggestions.This has been corrected.

  1. Line 348: „Song et al” should be cited „Song and Yu”

Reply:Thanks for your suggestions. This has been corrected.

  1. Line 539: Reference [37] - Please edit the species name with italic (Pinus sylvestris)

Reply: Thanks for your suggestions.This has been corrected.

  1. Line 551: Please check if these are authors last name: „Emily, G.; Jennifer, A.L.; Donald, R.S. Mechanisms contributing to stability in ecosystem function depend on the environmental context. Ecology Letters. 2010. 13, 1400-1410.” – they look like first names for me.

Reply: Thanks for your suggestions.This has been corrected.